# Tribological Properties of Nitrate Graphite Foils

**DOI:** 10.3390/nano14181499

**Published:** 2024-09-15

**Authors:** Nikolai S. Morozov, David V. Demchenko, Pavel O. Bukovsky, Anastasiya A. Yakovenko, Vladimir A. Shulyak, Alexandra V. Gracheva, Sergei N. Chebotarev, Irina G. Goryacheva, Viktor V. Avdeev

**Affiliations:** 1Lomonosov Moscow State University, Moscow 119991, Russia; demchenko.d@graflex.ru (D.V.D.); shulyak.v@unichimtek.ru (V.A.S.); gracheva.a@inumit.ru (A.V.G.); chebotarev.s@inumit.ru (S.N.C.); goryache@ipmnet.ru (I.G.G.); avdeev@highp.chem.msu.ru (V.V.A.); 2Ishlinsky Institute for Problems in Mechanics RAS, Moscow 119526, Russia; bukovskiypo@ipmnet.ru (P.O.B.); anastasiya.yakovenko@phystech.edu (A.A.Y.)

**Keywords:** graphite foil, roughness, coefficient of friction, mechanical properties, stress–strain state, structure, microstrains, residual macrostresses

## Abstract

This study investigates the tribological properties of graphite foils (GF) with densities of 1.0, 1.3, and 1.6 g/cm^3^, produced from purified natural graphite of different particle sizes (40–80 μm, 160–200 μm, >500 μm). Surface roughness was measured after cold rolling and friction testing at static (0.001 mm/s) and dynamic conditions (0.1 Hz and 1 Hz). Results showed that static friction tests yielded similar roughness values (*S_a_* ≈ 0.5–0.7 μm, *S_q_* ≈ 0.5–1.0 μm) across all densities and particle sizes. Dynamic friction tests revealed increased roughness (*S_a_* from 0.7 to 3.5 μm, *S_q_* from 1.0 to 6.0–7.0 μm). Friction coefficients (µ) decreased with higher sliding speeds, ranging from 0.22 to 0.13. GF with 40–80 μm particles had the lowest friction coefficient (µ = 0.13–0.15), while 160–200 μm particles had the highest (µ = 0.15–0.22). Density changes had minimal impact on friction for the 40–80 μm fraction but reduced friction for the 160–200 μm fraction. Young’s modulus increased with density and decreased with particle size, showing values from 127–274 MPa for 40–80 μm, 104–212 MPa for 160–200 μm, and 82–184 MPa for >500 μm. The stress–strain state in the graphite foil samples was simulated under normal and tangential loads. This makes it possible to investigate the effect of the anisotropy of the material on the stress concentration inside the sample, as well as to estimate the elasticity modulus under normal compression. Structural analyses indicated greater plastic deformation in GF with 40–80 μm particles, reducing coherent-scattering region size from 28 nm to 24 nm. GF samples from 160–200 μm and >500 μm fractions showed similar changes, expanding with density increase from 18 nm to 22 nm. Misorientation angles of GF nanocrystallites decreased from 30° to 27° along the rolling direction (RD). The coherent scattering regions of GF with 40–80 μm particles increased, but no significant changes in the coherent scattering regions were observed for the 160–200 μm and >500 μm fractions during dynamic friction tests. Microstrains and residual macrostresses in GF increased with density for all fractions, expanding under higher friction-induced loads. Higher values of both stresses indicate a higher level of accumulated deformation, which appears to be an additional factor affecting the samples during friction testing. This is reflected in the correlation of the results with the roughness and friction coefficient data of the tested samples.

## 1. Introduction

Graphite foil is a low-density graphite material [1,2] that has been incorporated into a vast scope of industrial applications [3,4]. Graphite foil (GF) is produced by rolling the exfoliated graphite without a binder of various densities for different ways of its application in nuclear, thermal energy, oil, and gas industries: sealing gaskets [5,6], flange gaskets [7,8], climate panels [9,10], heatsinks [11,12], etc. [13,14].

One of the significant problems with the flange and sealing joints [15,16] made of GF is leakages from the valve stem [17], which constitute a significant part of fugitive emissions released from the petrochemical and chemical plants (up to 50%) [18]. Since the world standards and legislation on fugitive emissions are becoming more and more strict, the methods, technologies, and materials applied for sealing joints shall be improved. Graphite sealing gaskets with soft packing are widely used as the sealing assemblies in various industry sectors [19,20]. Over recent years, owing to the intention to apply these seals under more constraint operating conditions [21,22], the recommendations were elaborated on how to improve the structural design of the sealing assemblies, as well as on how to use the novel composite materials developed on the basis of the thermally expanded graphite [23,24].

The end-use properties of the graphite seals, as well as their performance characteristics, are impacted by the tribological properties of the graphite foils [25], which are used in the manufacturing of the products [26]. The adhesive properties of the foil, which determine the tightness of the graphite seals, are presupposed by the quality of the foil surface. The initial roughness of the foil surface may have a significant effect on the wear resistance of the gasket and sliding joints [27]. Yet, such indicators have been insufficiently studied. The dynamic and static friction coefficients of the graphite foils may considerably impact the end-use properties as well [28]. For the purposes of researching the behavior of the sealing and flange gaskets under various operating modes, as well as in order to mitigate emergency failures as a consequence of the destruction of such gaskets, it is required to study the tribological properties of GF, which these products are made from. Currently, there is a noticeable gap in the literature concerning the study of tribological properties in the frictional interaction of nitrate graphite foils.

This research paper examines the friction coefficient and surface roughness of graphite foils (GFs) with densities of 1.0, 1.3, and 1.6 g/cm^3^. These foils were derived from natural purified graphite with particle sizes of 40–80, 160–200, and >500 μm. The novelty of this research lies in the study of the tribological properties of graphite foils. This characteristic of GF has not been addressed in scientific publications. The elastic characteristics of GFs were studied. The model for the stress–strain state of GFs under friction for the isotropic and anisotropic scenarios was built. The structural characteristics of the material were researched, and the microstrains inside the nanocrystallites, as well as the residual macrostresses in GFs, were determined. Additionally, understanding the nature and parameters of the tribological properties of graphite foils will help to further investigate issues related to the performance of graphite seals and flange gaskets.

## 2. Materials and Methods

### 2.1. Getting Materials for the Research Study

The natural graphite, with an average particle size of ~200 μm and a carbon content of >99.9% by weight, was sieved through a vibroshaker screen, with the sieves installed for the fractions of 40, 80, 160, 200, and 500 μm, which enabled isolation of the three target fractions being 40–80, 160–200, and >500 μm. The tribological properties of each batch of graphite foils were studied depending on the graphite fraction.

The intercalated graphite was prepared for each graphite fraction using the same method. The natural graphite powder was mixed with 98% nitric acid, a graphite-to-acid mass ratio of 1/0.8. The mixture was stirred for half an hour until spontaneous heating of the graphite ceased entirely. In order to obtain the oxidized graphite (OG), 1000 mL of distilled water was added to the graphite–acid mixture while being stirred, followed by transferring the resulting solution to the Nutsche filter and subsequently washed with 750 mL of distilled water. The resulting OG underwent drying in the drying oven for 6 h at a temperature of 50 °C until complete water evaporation.

Foaming of the oxidized graphite was carried out by its prompt-burst heating, reaching the temperature of 1000 °C in the air for ~3 s. The resulting powder of the thermally expanded graphite (TEG) for each lot weighing 33.75, 43.88, and 54.00 g got compressed in the 15 × 15 cm mold under the 2-ton loading in order to obtain the low-density GF-blanks, which were then rolled to reach the densities of 1.0, 1.3, and 1.6 g/cm^3^, respectively. Thus, three lots of graphite samples with different fractional compositions and three distinct densities were obtained. In total, 9 GF samples were made available.

### 2.2. Methodology for the Tribological Research Studies

Friction testing was conducted using the UMT-3MT tribometer (CETR, USA), involving the full-contact scheme with the mutual overlap coefficient being MOC = 1 (Figure 1). The static and dynamic friction coefficients were studied [29]. In order to determine the static friction coefficient, the transition point from the rest state to the sliding state was recorded for the coworking surfaces of GF and the counterbody. The static friction coefficient was measured at the low sliding speeds (not exceeding *V* = 1 μm/s) by means of the linear stepper motor. The dynamic friction coefficient was found under steady-state conditions, for which the constant value of the friction coefficient is pertained. The tests to determine the dynamic friction coefficient were carried out under the reciprocating motion conditions for the GF sample relative to the steel counterbody. The coefficient of friction, according to Amonton–Coulomb law, is defined as the ratio of tangential force to normal force. Moreover, Coulomb discovered that this coefficient also depends on the relative velocity of surfaces, duration of motion, and normal force. The frequency of the reciprocating motion was equal to *ω* = 0.1 and 1 Hz, and the movement amplitude was at 12.5 mm. The normal load (*F*) to determine both the static and dynamic friction coefficients was at *F* = 500 N. In order to obtain reliable results, the experiments were replicated three times, both for the purpose of determining the static and dynamic coefficients of friction for all the samples under investigation. Subsequently, employing the method of mean-square deviation, the mean values and the confidence intervals of the coefficients of friction were calculated.

The steel counterbody, designated as element 1 in Figure 1, was attached with the screws to a 2-component force sensor, onto which the normal load F was transmitted through the screw-type gear. The GF sample, identified as element 2 in Figure 1, was positioned within a holder, designated as element 3 in the same figure, that performed a reciprocating motion at the frequency ω owing to using the crank assembly, the rotational motion onto which was applied by the stepper motor. Rectangular steel 52,100 plates, measuring 60 mm in length, 30 mm in width, and 10 mm in height, were employed as the counterbodies. Prior to conducting tribological experiments, the working surface of the counterbodies was ground to achieve an average roughness of *S_a_* equal to 0.8 µm. Following each tribological test, the surface was cleansed of the deposited graphite film using isopropyl alcohol. Should any severe scratches or other forms of damage occur on the counterbodies surface, it was subjected to further sanding.

### 2.3. Method of Measuring the GF Surface Roughness

The surface topography of the GF samples was studied with the involvement of the 3D non-contact profilometer S Neox (Sensofar, Barcelona, Spain), fitted with the motorized X-Y stage and the 10× confocal object lens. The single-scan area was 1.7 × 1.4 mm, with the 1224 × 1024 pixels resolution. Merging 136 nearby scans enabled obtaining the image of the surface roughness for the sample of 25 × 8 mm, which clearly demonstrated the surface roughness for the sample. Assessment of the roughness parameters was conducted pursuant to the ISO 25178 standard [30], which allows for the three-dimensional (by area) determination of the height and hybrid relief parameters using the device software tools. The software processing of the three-dimensional images was carried out according to the methodology described in the paper [31].

### 2.4. Determination of the GF Elastic Characteristics

The experimental data generated using the method of instrumental indentation were taken to compute the local modulus of elasticity using the NanoScan-4D scanning nano-hardness tester (Federal State Budgetary Institution TISNUM, Moscow, Russia). This methodology enables obtaining the load-to-depth relation when penetrating with the hard tip into the studied material in the course of loading and unloading with a high level of accuracy within the micrometer range [32,33]. In this research study, the ceramic (Al_2_O_3_) ball with a diameter of 2.1 mm was used, which was pressed into the samples applying the loads of *F* = 50, 100, 200, and 400 mN. Five repetitions took place for each load in different surface areas.

In order to assess the compliance for compression for the entire sample, singular load-unload testings were conducted all over the GF surface area. At that, the load increased in steps from 0 to 500 N at 25 N intervals. It needs to be noted that, to eliminate the friction impact in the course of loading, the sample surface was purposefully water-lubricated.

### 2.5. Scanning Electron Microscopy of the GF Samples

The GF microstructure was studied by means of scanning electron microscopy using the TESCAN VEGA 3 (Brno, Czech Republic). The electron gun of the device contained a lanthanum hexaboride cathode. The electron beam was assembled under the accelerating voltage of *HV* = 15 kV. The electron detector SE was used to register the secondary electrons. The SEM images featured the field of view being 750 × 750 μm.

### 2.6. XRD-Analysis of GF

#### 2.6.1. Structural Characteristics and Microstrains in the GF Nanocrystallites

The GF structural characteristics were studied involving the method of XRD analysis and using the Rigaku Ultima IV diffractometer (Tokyo, Japan) with Bragg–Brentano focusing geometry. Radiation is *CuK_α_* = 0.15418 nm. The XRD patterns for the GF samples were recorded in the *θ*/*2θ* mode of the diffractometer at the Bragg angles of *2θ* = 20–90°. The XRD pattern of the standard *LaB_6_* sample was recorded under similar conditions to account for the instrumental broadening. 

For the purposes of assessing the structural changes in graphite, the characteristics of the Bragg reflection in the XRD patterns were analyzed. In the crystalline samples that are under some mechanical or thermal treatment, the full width at half maximum (*FWHM*) represents the convolution of the effects from various components (Equation (1)). This material experiences external strains owing to cold rolling, so the computation of the coherent scattering region was carried out using the Williamson–Hall method, which allows for the impact from the microstrains (Equation (2)) [33,34,35]:(1)$$FWHM=\beta_{L}+\beta_{\varepsilon }+\beta_{instr};$$
(2)$$\beta_{phys}=FWHM-\beta_{instr}=\beta_{L}+\beta_{\varepsilon }=\frac{0.9\cdot \lambda }{L_{c}\cdot \cos\theta }+4\cdot \varepsilon \cdot \tan\theta ;$$
(3)$$\beta_{phys}\cdot \cos\theta =\frac{0.9\cdot \lambda }{L_{c}}+4\cdot \varepsilon \cdot \sin\theta ;$$
where *FWHM* is the full width at half maximum of the reflection in the experimental XRD pattern; $\beta_{phys}$ is the physical broadening of the reflection by the structural specifics of the material; $\beta_{instr}$ is the instrumental broadening by the geometry and diffractometer characteristics; $\beta_{L}$ is the X-ray reflection broadening by the nanocrystallites dispersibility; $\beta_{\varepsilon }$ is the input of the reflection broadening by the presence of the microstrains; *λ* is the X-ray radiation wavelength; $L_{c}$ is the size of the coherent scattering regions along the lattice parameter *c* of the graphite nanocrystal; *θ* is the Bragg angle; *ε* is microstrains in the nanocrystallites that appeared as the result of the mechanical processing.

Computation of the coherent scattering region of nanocrystallites was conducted along the parameter *c* of the nanocrystal grid as per the reflections (00.l). The relation of ${(\beta }_{phys}\cdot \cos\left(\theta \right))$ to (4·sin⁡θ) (Equation (3)) was elaborated and was approximated by the linear function and extrapolated up to the intersection with the axis of ordinates. The microstrains in the nanocrystallites were determined by means of the slope ratio, while the value of the coherent-scattering regions was determined by the value at the intersection point of the axis of ordinates and the trend line.

#### 2.6.2. Misorientation Angles of the Nanocrystallites

The misorientation angles (*MA*) of the nanocrystallites were determined by the rocking curves [36] for the reflection of the hexagonal graphite [00.6] on the Rigaku Ultima IV diffractometer and using the *CuK_α_* radiation at the Bragg–Brentano focusing in the diffractometer *θ*-mode when scanning along *θ* from 10° to 80°. The analytically designed *FWHM* for the resulting curves was considered the *MA* value.

#### 2.6.3. Residual Macrostrains

In order to determine the residual strains, the standard X-ray diffractometry method *sin*^2^*ψ* (Equation (4)) was taken. The underlying assumption of the method constitutes that relaxation of the normal stress occurs within the thin surface layer of the sample, and thus, the latter is in a plane stress state [37,38]. Through recording the reflection [00.6] of the sample tilt at certain angles *ψ*, it appears possible to record its stress state (Equation (4)).
(4)$$\varepsilon_{\varphi \psi }=a+{b\cdot sin}^{2}\psi =\frac{\Delta d}{d};$$
(5)$$a=-\frac{\nu }{E}\cdot \left(\sigma_{RD}+\sigma_{TD}\right);$$
(6)$$b=\frac{1+\nu }{E}\cdot \left(\sigma_{RD}{\cdot cos}^{2}\varphi +\sigma_{TD}\cdot {sin}^{2}\varphi \right);$$
where *ψ* is the angle of the sample tilt; *φ* is the angle between the direction of rolling and the direction of the X-ray beam; $\varepsilon_{\varphi \psi }$ is the strain at the tilt angle *ψ* in the direction as chosen; *ν* is the Poisson’s ratio for the material; *E* is Young’s modulus for the material; $\sigma_{RD}$ and $\sigma_{TD}$ are the residual macrostrain values along and across the direction of rolling; *d* is the value of the interplanar distance at the angle of the sample tilt *ψ*; Δ*d* is the change in the interplanar distance at the angle of the sample tilt ψ relative to the initial positioning at *ψ* = 0°.

In the course of this research study, measurement of the samples was conducted at the angles *φ* = 0° and 90°, at which $\sigma_{RD}$ and $\sigma_{TD}$, respectively, were measured.

### 2.7. The Stress–Strain State Model of the Loaded Sample

When proceeding with the numerical modeling of the stress–strain state of the samples, being the pressed layers of foil and having the shape of the bars of certain sizes, loaded with the normal and tangential forces, it was assumed that the samples themselves could be considered as a homogeneous continuous medium, i.e., the structure of the material is not taken into account. In order to assess the stresses and strains at each point of the sample under load, the stress tensors $\sigma_{ij}\left(x,y,z\right)$ and strain tensors $\varepsilon_{ij}\left(x,y,z\right)$ were used, where *i* and *j* are the directions of the tensor axes in the Cartesian coordinate system. The process of loading was considered isothermal. When setting the contact problem, two types of boundary conditions were taken into account: corresponding to the normal compression experiments and to its friction interaction.

#### 2.7.1. Normal Compression Modelling for the Sample

In order to simulate the stress state of the sample compressed along the *z*-axis (Figure 2), the experimental conditions described in Section 2.4 were applied. In the course of the normal loading experiment, the sample under study is compressed along the normal direction by another bar made of a more rigid material, which is loaded with a gradually increasing load *P*. On the basis of the vertical displacement measurements $u_{z}$ of the upper facet of the sample, given the fixed values of the monotonously increasing load, the relation $u_{z}\;(P)$ is plotted. When proceeding with modeling, it was assumed that no friction forces were in action in the area of the contact interaction, which was in line with the experiments on the water-lubricated contact surfaces. The boundary conditions as follows were adopted:lower facet: $u_{x}=u_{y}=u_{z}=0$;upper facet: $u_{x}=const,\tau_{xz}=\tau_{yz}=0$;with other facets being free from the loads. In order to plot the relation $u_{z}\;(P)$, the displacement $u_{z}$ is represented parametrically at 0.035 mm intervals, within the range from 0 to 0.35 mm, which corresponds to the outcome of the experiment.

In numerical computations, the compressible material of the sample was assumed to be homogeneous, isotropic, and elastic. Owing to the finite deformations of the sample, the linear theory of elasticity is not applicable. Therefore, in order to describe the mechanical behavior of the material, the hyperelastic model was used, namely a neo-Hookean solid, for which two material constants are sufficient: Young’s modulus (*E*) and the Poisson’s ratio (*ν*). At that, the function of the energy density in elastic strain *W* is determined by the expression [39]:(7)$$W=\frac{\mu }{2}\left(I_{1}-3-2ln\left(J\right)\right)+\frac{\lambda }{2}{\left(ln\left(J\right)\right)}^{2};\;J=det(F)$$
where $I_{1}$ is the first invariant of the right Cauchy–Green strain tensor C; *F* is the strain gradient tensor; *λ* and *μ* are the Lamé elastic constants. Equation (7) is then used to find the relationship between stresses and strains (Equation (8)).
(8)$$S=2\frac{\partial W}{\partial C};$$
where *S* is the second Piola–Kirchhoff stress tensor. The Lamé elastic constants present in Equation (7) are linked with Young’s modulus *E* and Poisson’s ratio *ν* by the relations (9).
(9)$$\lambda =\frac{\nu E}{\left(1+\nu \right)\left(1-2\nu \right)};\;\mu =\frac{E}{2\left(1+\nu \right)}.$$

The problem was solved using the finite element method (FEM), which is widespread in the research studies of the stress–strain state of the bodies under large strains or in the case of materials featuring inelastic properties [40], i.e., in the case of the nonlinear problems occurring when dealing with the finite strains of the bodies. The problem was resolved in the spatial setting: the bar has a length of 24 mm, its width is 7.8 mm, and its height is 1.4 mm (the size of the samples used in the compression experiment). The grid was built uniformly, and the tetrahedrons were used as the finite elements (Figure 2). The nodal values of the displacement components $u_{i}$ are derived from the solution of the discretized boundary value problem, while inside the elements, these values are approximated by certain functions, such as the Lagrange polynomials at the given set of data. In order to determine the stress state of the material, the variational approach based on the principle of probable displacements was used.

Given that the experimental curves are quite near to each other, just one of them was taken into consideration, and the values of the constants *E* and *ν* were selected in such a way that the curve plotted on the basis of the numerical computation would be close to the data of the tests (see Section 2.4). In particular, for the values of the Poisson’s ratio, *ν* = 0.1 and *ν* = 0.3, Young’s modulus *E* varies at 0.5 MPa intervals.

#### 2.7.2. Modelling of the Frictional Contact of the Sample

In the friction experiment described in Section 2.2, the bar is first compressed by another more rigid bar with the given normal load *P*, followed by the reciprocating movement of the rigid bar. At that, the tangential force gets settled, and the friction coefficient *f* is determined based on Coulomb’s law of friction states. In particular, some part of the sample is in the agile tooling, meaning its movements are limited.

For the purposes of simplifying the numerical computations and improving the precision of the results, the friction is neglected at the normal load stage, and the vertical displacement $u_{z}$ is specified, yet not the normal load *P*. The boundary conditions are as follows:lower facet: $u_{x}=u_{y}=u_{z}=0$;upper facet: $u_{z}=const,$ $\tau_{yz}=-f\sigma_{z}$, $\tau_{xz}=0$;

On the side facets, except for the top layer of 1 mm thick, which is not in the agile tooling and the chamfers, the zero-displacement in the normal line direction ($u_{n}=0$) to the surface is imposed; the remaining parts of the side wall and chamfers are free from any loads. This setting of the problem presupposes that the upper and more rigid bar slides along the sample (the upper bar gets displaced along the axis *z* by the value sufficient to have *P* = 500 N and along the axis *y* (Figure 1 and Figure 3), while the Coulomb’s law is fulfilled in the contact area [41], with the friction coefficient being *f* = 0.2 (average value of the experimental results).

The material for both bars is assumed to be homogeneous and linearly elastic, with the Poisson’s ratio being equal to 0.3. At that, the rigid bar is assumed to be isotropic, with the Young’s modulus being equal to 1000 GPa (which is much higher than the Young’s modulus of the sample material). Two cases are studied for the sample when it is modeled either as the isotropic or the anisotropic (transversely isotropic) linearly elastic body. Hooke’s law, proper for the linear theory of elasticity, can be written in matrix form as follows for the case of the transversely isotropic body (Equation (10)).
(10)$$\left[\begin{array}{c} \begin{array}{c} \begin{array}{c} \begin{array}{c} \begin{array}{c} \sigma_{x}\\ \sigma_{y} \end{array}\\ \sigma_{z} \end{array}\\ \tau_{yz} \end{array}\\ \tau_{xz} \end{array}\\ \tau_{xy} \end{array}\right]=\left[\begin{array}{c} \begin{array}{c} \begin{array}{c} \begin{array}{c} \begin{array}{c} C_{11}{\;C}_{12}\;C_{13}\;0\;0\;0\\ C_{12}{\;C}_{22}{\;C}_{13}\;0\;0\;0 \end{array}\\ C_{13}{\;C}_{13}{\;C}_{33}\;0\;0\;0 \end{array}\\ \;0\;0\;0\;C_{44}\;0\;0 \end{array}\\ \;0\;0\;0\;0\;C_{44}\;0 \end{array}\\ \;0\;0\;0\;0\;0\;\frac{C_{11}-C_{12}}{2} \end{array}\right]\left[\begin{array}{c} \varepsilon_{x}\\ \varepsilon_{y}\\ \varepsilon_{z}\\ \gamma_{yz}\\ \gamma_{xz}\\ \gamma_{xy} \end{array}\right].$$

The five independent constants comprising Equation (10) are related to the prominent engineering moduli of elasticity (Equation (11)).
(11)$$E_{x}=E_{y}=\frac{\left(C_{11}-C_{12}\right)\left(C_{33}\left(C_{11}+C_{12}\right)-2C_{13}^{2}\right)}{C_{11}C_{33}-C_{13}^{2}};\;E_{z}=\frac{C_{33}\left(C_{11}+C_{12}\right)-2C_{13}^{2}}{C_{11}+C_{12}};\;\;\nu_{xy}=\nu_{yx}=\frac{C_{12}C_{33}-C_{13}^{2}}{C_{11}C_{33}-C_{13}^{2}};\;\nu_{zx}=\nu_{zy}=\frac{C_{13}}{C_{11}+C_{12}};\;G_{xz}=G_{yz}=C_{44}.$$

In Equation (11), the first index in Poisson’s ratios signifies the main longitudinal strain (extension) axis, while the second is the axis of transverse strain (contraction). The remaining parameters, i.e., $G_{xy}$, $\nu_{xz}$, and $\nu_{yz}$, are determined by the already written parameters (Equation (12)).
(12)$$G_{xy}=\frac{E_{x}}{2\left(1+\nu_{xy}\right)};\;\nu_{xz}=\nu_{yz}=\frac{E_{x}}{E_{z}}\nu_{zx}.$$

For the anisotropic case, the elasticity tensor components are set as follows [42]: *C*_11_ = 1060 GPa, *C*_12_ = 180 GPa, *C*_13_ = 15 GPa, *C*_33_ = 36.5 Gpa, and *C*_44_ = 0.18 GPa, which are typical for the graphite crystal. For the isotropic case, the Young’s modulus is taken as equal to the Young’s modulus along the *O_z_* axis in the anisotropic case, i.e., $E=E_{z}\approx 36\;GPa$ and the Poisson’s ratio is *ν* = 0.3.

The solution to the problem was also found through the finite element method, which is efficient for such problems when the friction of the bar element is being studied [43,44,45]. As for the numerical computation, the model of the sample is built (length is 24 mm, width is 7.8 mm, height is 2.5 mm), as well as the model of the counterbody (length is 74 mm, width is 30 mm, and height is 10 mm), with the finite elements of which being tetrahedrons (Figure 3). In the sample’s model, the sharp edges get smoothed by adding the chamfers (the upper and side facets), with a radius of 0.1 mm. The counterbody mesh is uniform, but it gets thicker for the sample near the boundaries.

The research study of the components of the internal stress tensor has great practical significance because it enables the determination of plastic deformations and crack initiations. Note that in this case, the surface contacting bodies’ roughness, which impacts the distribution of the stresses near the contact region [46] and also influences the contact rigidity [47], is not taken into consideration. With that in mind, it is required formulation and solution of contact problems with taking into account both the macro- and microgeometry of the interacting bodies for the benefit of a more accurate analysis.

## 3. Results and Discussion

### 3.1. Research Study of the GF Surface Roughness

The original GF surface quality was studied using the 3D profilograms (Figure 4). For all 3D surface analyses, the influence of edge effects that were introduced during the preparation of the GF samples was excluded. Additionally, rolling traces were observed on the surfaces of the original GF samples. These traces are characterized by localized gradients in surface roughness.

Figure 5 demonstrates the surface 3D profilograms of graphite foils from the 40–80 μm fraction after tribological testing. Figure 6 presents the relationships between the arithmetic mean roughness (*S_a_*), the root mean square roughness (*S_q_*), and the density of the graphite foils (GF). Similar 3D profilograms were obtained for all fractions and provided in the Appendix A.

Following the testings for the dynamic coefficient of friction, the GF surface from the 40–80 μm fraction turned out to be more developed, which is particularly prominent at the frequency of the reciprocating movement of 1 Hz. Additionally, the traces of partial (Figure 5f) or stronger (Figure 5i) delamination of GF for the densities of 1.3–1.6 g/cm^3^. In Figure 5, delamination is represented by a sharp gradient in the color palette. For example, in a GF sample with a density of 1.6 g/cm^3^ (Figure 5i), delamination occurred without separation of part of the layer, as illustrated in Figure 5d,e. This sample exhibits internal delamination, evident by a smoother color gradient. The blue color in the figure indicates that this portion of the sample remains, while everything else has detached the inner layers but has not extended beyond the friction zone. The samples of the lower density are not so significantly distorted, and that is also evident as per the roughness parameters (Figure 6a).

The GF samples were tested for the dynamic coefficient of friction, revealing traces of tearing out/delamination of some foil layers, possibly owing to low adhesion between them. In this case, the surface becomes more developed (a larger number of peaks and valleys are observed), regardless of the density of the sample.

The surface of the GF samples obtained from the >500 μm fraction, following the experimental study of the dynamic coefficient of friction, exhibited unidirectional friction traces. Additionally, it showed minor signs of wear and destruction. The surfaces primarily contain traces of abrasive wear, i.e., the areas of delamination or tear-out of the layers of the material are basically non-existent.

The profilograms from the static friction experiments showed that the GF surfaces were very similar in shape to their original surfaces (Figure 4). Only sporadic traces of bearing strain or wear were observed. As the fraction size in the GF composition increased, the roughness of the original surface also increased (Figure 6). The density of the GF had a negligible impact on the initial roughness. The minimum initial roughness, regardless of density, was observed on the surfaces of the GF samples from the 40–80 μm fraction, with values ranging from *S_a_* = 0.41 to 0.48 μm.

The measured-data analysis for the roughness parameter (Figure 6) demonstrated that, when tested at low sliding speeds (static friction coefficient), the height parameters of roughness differ insignificantly from the original values (Figure 6, the black lines). With the rising oscillation frequency of the linear drive, the roughness values increased as well. For instance, for the GF sample with a density of 1.3 g/cm^3^ and from the fraction of >500 μm, the roughness rose by around 4-fold (*S_a_* = 6.7 μm) if compared to the original surface (*S_a_* = 1.5 μm). This was associated with strong abrasive wear.

The roughest surface was demonstrated by the GF samples from the 160–200 μm fraction. The *S_a_* parameter for these samples, regardless of density, ranged from 0.58 to 4.09 μm. This increased roughness is attributed to significant delamination of the samples, which indicates poor interlayer adhesion.

### 3.2. Friction Coefficient of Graphite Foils

Figure 7 shows typical graphs for recording static (Figure 7a) and dynamic (Figure 7b) friction coefficients. The static coefficient of friction was defined as the maximum peak, which corresponds to the moment of transition from a state of rest to the movement of touching bodies. There are two distinctive zones on the graph of the dynamic coefficient of friction: the run-in zone and the steady-state friction zone. All carbon materials have a specific feature of graphite film formation; therefore, until a sufficiently thick film is formed on the counterbody, the dynamic coefficient will be high [48].

Figure 8 demonstrates the results of the GF sample testing, revealing to what extent the increase in the sample density impacts the friction coefficient (static and dynamic).

The measured-data analysis demonstrates that the static friction coefficient decreases with the increase in density of the GF samples by all fractions. This is associated with its more fragile behavior in the tangential direction. The lowest dynamic friction coefficient pertained to 40–80 μm fraction of GF and equals to *µ* = 0.13 ÷ 0.15. At that, the highest values of the friction coefficient pertain to the GF sample with the fraction size of 160–200 μm (*µ* ≈ 0.21), which is largely because of the great destruction of its surface (Figure 4). The dynamic friction coefficient exhibits weak dependencies on the GF density, with the predominant factor in these experiments being oscillations of the linear drive. For a GF sample with a density of 1 g/cm^3^ and a fraction size of 160–200 μm, a pronounced stratification of the material was observed under conditions of dynamic friction at a frequency of 1 Hz. In this context, friction manifested itself as a phenomenon occurring between two distinct layers of the GF, with one layer remaining within the contact area and undergoing oscillatory movements while the other adhered to the steel counterbody. The primary mechanism underlying this phenomenon appears to be the presence of weak adhesive forces between individual layers of the GF material. At lower sliding velocities for this particular sample, the separated particles of GF were observed to freely exit the contact region, resulting in friction between the GF and the steel counterbody and, consequently, a higher coefficient of friction.

The marked disparities in the measured coefficients of friction for materials with varying densities and fraction sizes underscore the critical role of surface imperfections in the context of friction. When GF interacts with a smooth and hard surface, it gives rise to shear deformations, which constitute a pivotal aspect of the friction phenomenon. This notion was also underscored by Warburton in his seminal work [49]. In the literature, there is an analysis of the process of formation of a so-called “third body”, which can occur between two sliding surfaces [48]. These are structures that form on one or both surfaces of the element, creating a friction pair. They are typically composed of broken graphite particles. During the movement of the elements relative to each other, these structures can be formed and removed from the surface. This process of formation and removal of these structures leads to the destruction and reformation of the friction layer.

### 3.3. Determination of the Elastic Characteristics of GF

Figure 9 demonstrates a typical diagram for loading–unloading in relation to the depth of penetration obtained in the course of the instrumental indentation. This result exhibits the occurrence of plastic and residual deformations after indentation, which can be detected on the unloading curve.

Using the software assigned to the nanohardness tester, computation of the elastic modulus was conducted (Figure 9b). Based on the outcome of the Figure 9b analysis, it can be concluded that, with increasing the fraction size in the GF composition, the local elastic modulus decreases, which is associated with the weak mechanical properties of any separate carbon grain. An increase in density implies that the interlayer strength is rising, owing to which the elastic modulus becomes higher (Figure 9b).

Figure 9c demonstrates the dependence of the compression depth for the GF sample on the applied load. For all curves (Figure 9c), there is a noticeable transition from the zone of small deformations to the zone of plastic deformations. This transition occurs approximately at an implementation load of 100 N. In this case, when the load is removed, a slight recovery of the sample will occur, when most of it will be subject to irreversible plastic deformation. Similarly to the instrumental indentation, the samples with a fractional composition >500 reveal the largest compression depths, i.e., the lowest elastic modulus.

### 3.4. SEM Study of the GF Samples

The electronic microscopy was helpful in obtaining the images for GF from the 40–80 μm fraction before (Figure 10a) and after the testings of static (Figure 10b) and dynamic (Figure 10c,d) frictions.

Based on the outcome of the image analysis, it can be concluded that with the intensification of the mechanical load on the surface of the graphite foil (GF), the appearance of visible defects becomes increasingly apparent. This process leads to the destruction and delamination of the foil, which is most prominently observed in the friction testing results at frequencies of 0.1 and 1.0 Hz. Notably, during dynamic friction testing, a significant amount of delaminated graphite particles was observed. At higher frequencies, these particles were smaller in size, likely due to the additional breakage of the delaminated particles under higher frequency loads. These findings significantly impact the tribological properties of the foil, confirming the previously obtained results. The observed defects and structural failures under enhanced mechanical stress suggest a direct correlation between the applied load and the degradation of the material’s integrity.

### 3.5. Stress–Strain State in the Graphite Foils

When proceeding with the computations as per the linear theory of elasticity (Figure 11a), the obtained numerical results differ not only quantitatively but also qualitatively from the experimental results. The Young’s modulus in the numerical computations was taken as equal to 10 MPa (*ν* = 0.1 for the red line and *ν* = 0.3 for the blue line). In order to bring the numerical computation more in concordance with the experimental results, the hyperplastic model was taken in lieu of the linear model, namely the neo-Hookean solid (1), which is also determined by two parameters (Figure 11b,c). The same model was studied in the paper [50] for modeling the strains in the graphite sealing rings.

The minimum absolute error (difference between the experimental and model values) in the case of the material with *ν* = 0.1 is attained at *E* = 8.5 MPa and in the case of the material with *ν* = 0.3 at *E* = 7 MPa. In both cases, the absolute error, when evaluating the displacements, does not exceed 22 μm for the experiment under consideration within the load range from 0 to 500 N. It follows from the analysis of the numerical results that the Poisson’s ratio bears no qualitative impact on the type of dependence $u_{z}\;(P)$, yet only quantitatively. However, the strained state of the sample depends on the compression of the material (Figure 12a). The visual inspection of the sample after the applied strain allows us to make assumptions about its compressibility. Still, in any case, Young’s modulus of the material has values near 10 MPa. This value is 10-fold lower than what was obtained by means of the indentation method; this implies significant differences in the local characteristics from the volumetric ones, which can be explained by the heterogeneity of the sample material.

The stress–strain states for the isotropic and anisotropic models obtained in the course of numerical computations differ, both quantitatively and qualitatively. Thus, the 500 N load corresponds to the vertical displacements of 0.144 μm for the isotropic model and 0.186 μm for the anisotropic model. The maximum values of the von Mises stress distribution in the studied sample over the entire volume under normal load differ as well: 9.21 MPa for the isotropic model and 22.17 MPa for the anisotropic model. After the shear, the maximum value of the von Mises stress in the isotropic bar is at 17.91 MPa and in the anisotropic bar at 925.78 MPa. In both cases, the maximum values of the von Mises stress distribution are attained near the edge of the agile tooling, which is the beginning of the free part of the sample not constrained by the tooling.

The most striking difference in the results is observed when studying the shear stresses, namely the stress tensor component $\tau_{yz}$ in the middle plane *O_yz_*. As opposed to the von Mises stresses, the shear stresses shall be lower in the case of the anisotropic model. Thus, the maximum values of the tangential component of the stress tensor under normal load are 3.57 MPa for the isotropic bar and 0.32 MPa for the anisotropic bar. After the shearing, the maximum value in the isotropic bar is 6.97 MPa and 0.66 MPa in the anisotropic bar. Figure 12 demonstrates the shear stress distribution in the bar made of isotropic (Figure 12c) and anisotropic (Figure 12d) materials in the *O_yz_* (x = 0) after the shear. The results make it possible to conclude that in the case of the isotropic model, the maximum values shall be near the edge of the agile tooling and in the case of the anisotropic model near the upper facet of the sample.

### 3.6. Structural Characteristics of GF

#### 3.6.1. Structural Changes in GF

The research study of the GF structural characteristics was carried out in relation to the type of frictional action (Figure 13 and Figure 14). The sizes of the coherent-scattering region (i.e., their dimensions) for the nanocrystallites (Figure 13) were studied by means of the Williamson–Hall method using the XRD patterns, as initially GF cold rolling.

The measured-data analysis demonstrates that the nanocrystallite sizes of the foils by the fraction of 40–80 μm (Figure 13a) are larger than those from others. This deviation may stem from two facts: the nature of the original graphite, i.e., the smaller particles have larger nanocrystallites, or the difference in the intercalation and foaming processes of graphite, resulting in less destruction of the crystallites during foaming of the particles in this fraction is significantly lower than in other fractions. The original material by the fraction of the 40–80 μm (the black line) reveals the increase in the coherent-scattering region plateauing at a density greater than 1.3 g/cm^3^. This may result in the fact that during X-ray radiography, the larger particles and more deformed particles come into the reflective position due to other factors not taken into account in the Williamson–Hall method. At that, for the fraction of 40–80 μm, the increase in the size of the nanocrystallites from 21 to 28 nm was observed in the foil with the density of 1.0 g/cm^3^ as the load on the samples increased.

Under static friction, no changes occurred in the crystallite sizes, and consequently, the applied load was not sufficient to trigger the rotation of the graphite particles or plastic deformation of the nanocrystallites. For the samples under heavy load (the green and red graphs), the decrease in the coherent-scattering region takes place from 28 to 24 nm as the density increases. This trend was attributed to the fact that with the increasing density, the larger number of particles began to interact with each other under friction, which caused the increase in plastic deformations within the volume of the material, which in turn resulted in the decreasing size of the nanocrystallites.

A different pattern was observed for other fractions. For the original and static friction-induced GF by the fractions of 160–200 μm and >500 μm (Figure 13b,c the black and blue lines), a similar increase in the coherent-scattering region growth was observed, similar to the sample of the original GF by the fraction of 40–80 μm (Figure 13a, the black line), due to the same reasons. The stability of the coherent-scattering region for friction GF (Figure 13b,c the green and red lines) was due to the action of two factors: repositioning of the graphite particles, which led to the coherent-scattering region growth, and plastic deformation under friction, which causes the coherent-scattering region reduction, i.e., the effects neutralized each other.

A similar pattern is observed when examining the misorientation angles in GF in the rolling direction (RD) (Figure 14a–c). For GF across all fractions, both the original material and the samples subjected to static friction exhibit a consistent trend: the misorientation angles decrease with increasing density. This phenomenon is attributed to the repositioning of graphite particles during rolling (Figure 14a–c the black and blue lines). For the foils from the 40–80 μm fraction, a reduction in misorientation angles is observed when transitioning from a density of 1.0 to 1.3 g/cm^3^. This suggests that the plastic deformations induced by friction have a minimal effect at these densities. However, at a GF density of 1.6 g/cm^3^, an increase in misorientation angles occurs due to the heightened plastic deformation. This trend is also evident in GF from the 160–200 μm and >500 μm fractions, though with a notable difference: under more intense mechanical action resulting from friction, the plastic deformation mechanism for the nanocrystallites begins to dominate at densities as low as 1.3 g/cm^3^ (Figure 14a–c, the red lines).

Destruction is significantly lower in the transverse direction, as evidenced by the uniform distribution of misorientation angles across all fractions. This observation arises because the reciprocating motions of the graphite foil during the tribological testing were applied along the RD and were not examined in the transverse direction (TD).

#### 3.6.2. Microstrains and Residual Macrostresses in GF

The Williamson–Hall method enables evaluation of the inner distortions of the nanocrystallites appearing due to rolling. Such distortions are easy to represent through the microstrains within the crystal grid of the crystallites. The results for GF with the density of 1.0, 1.3, and 1.6 g/cm^3^ by the fractions of 40–80 μm, 160–200 μm, and >500 μm are shown in Figure 15.

For the original GF by the 40–80 μm fraction, the increase in microstrains is observed due to rolling (Figure 15, the black line), with the increasing density followed by reaching the plateau. This pattern is similar to the coherent-scattering region distribution in GF from this fraction and is justified by the same causes. As for the original GF from other fractions, the value of the microstrains increases on a constant basis (Figure 15b,c, the black line), which is also consistent with the results for the coherent-scattering region size and is justified by the same causes.

As for the GF from the fraction of 160–200 µm and >500 µm, when changing the density, the microstrains do not differ from the original GF. This implies that when these fractions are under static friction, no significant structural changes occur in them, i.e., no plastic deformation of the samples.

Still, no changes occur for GF from the 40–80 μm fraction (Figure 15a) in view of the microstrains under static friction with the change in density, despite the fact that for the original foil, the increase in strains is observed. So, in GF of the finely dispersed particles, under static friction, the plastic deformations take place, during which relaxation of the microstresses occurs. This is similar to GF as well when studying friction at the speed of 0.1 Hz, with the only difference being that more accumulated strain pertains to the lower densities.

For GF, at the friction of 1.0 Hz, relaxation of the microstresses is spotted, obviously caused by the plastic deformation and destruction of the foil. For the GF samples from the 160–200 μm fraction (Figure 15b), a similar scenario takes place in the course of the friction testings, as with the original material, with only a slight increase in the degree of strain of the crystallites because of the mechanical action.

As for the GF samples by the >500 μm fraction (Figure 15c), the distribution pattern similar to the original one is observed the same as for the original sample; yet, as seen in the figure, the friction causes the increase in microstrains of the nanocrystallites in the larger fraction because of the plastic deformation.

The residual macrostresses were studied for GF as well, caused by the effect of the nanocrystallites on each other and the porous structure of GF. The results of the studies are shown in Figure 16. A similar situation pertains to all samples: the residual stresses for the initial samples are at their minimum levels relative to the tested ones and decrease slightly with increasing density. In this case, the decrease is more pronounced with the increasing particle size of graphite. With the increasing density and increasing the mechanical load on the samples, the increase in the residual macrostresses is observed, which is virtually the same for all fractions. Apparently, the residual stresses, to a greater extent, are impacted by the preload applied to the samples rather than by the mechanical load. One value of the residual macrostresses in GF decreases with the increase in fraction, which is reasonable, as there are fewer interactions between the large-sized particles owing to their lower amount.

Along the cross-section, for the GF samples of all fractions (Figure 16d–f), no particular changes are recorded in the residual stresses. This is due to the fact that the destruction of the foils occurred along the rolling direction, yet in the transverse direction, there were no significant changes. The deviations in the residual stresses of the GF with a density of 1.0 g/cm^3^ from the 40–80 μm fraction during friction testing at 0.1 Hz are attributed to the fact that this sample experienced more significant wear compared to others (Figure 16d, green line). This effect is likely due to local inhomogeneities in the material thickness that developed during rolling. This is evident from the fact that the residual stress values for the other densities of this foil fraction correlate with the data obtained under different friction testing conditions. The runouts, as shown in the graphs, are explained by the fact that, when measuring the residual stresses in the transverse direction, it was impossible to avoid the impact from the edge effects of the samples.

Nonetheless, the magnitude of the residual macrostresses is large enough for the graphite foils, the tensile strength of which varies from 6 to 15 MPa. Consequently, the presence thereof may impact differently on the foil’s properties, the mechanical and tribological ones in particular.

Higher values of both stresses indicate a higher level of accumulated deformation, suggesting that this deformation is an additional factor influencing the failure of the samples during friction testing. This is evident from the correlation observed between the increased stress values and the corresponding data on surface roughness and friction coefficients of the tested samples. The elevated stress levels contribute to changes in the microstructure and mechanical properties of the graphite foils, thereby impacting their tribological performance. Consequently, understanding this relationship is crucial for accurately predicting the behavior of graphite foils under operational conditions. Further investigation into this phenomenon will enhance our ability to design and optimize graphite-based materials for various industrial applications.

## 4. Conclusions

Based on the analysis of the tribological properties of GF from different fractions, it can be concluded that the surface roughness values remain unchanged (*S_a_* ≈ 0.5–0.7 μm, *S_q_* ≈ 0.5–1.0 μm) with the density decreasing from 1.0 to 1.6 g/cm^3^ for both the original material and after static friction. The friction during cold rolling had a similar effect on the structural changes in the GF samples, and the weak loads used in determining static friction did not cause significant surface defects.

For GF subjected to dynamic friction testing, an increase in roughness (*S_a_* from 0.7 μm to 3.5 μm and *S_q_* from 1.0 μm to 7.0 μm) is observed with the density increase from 1.0 to 1.6 g/cm^3^ for each fraction. The GF friction coefficient increases with the fraction size from 0.15 to 0.167, particularly at a density of 1.6 g/cm^3,^ and decreases from 0.165 to 0.15 with the density increase within each fraction.

Mechanical testing using the instrumental indentation method revealed that the GF elastic modulus increases with density regardless of fraction size. However, with an increase in fraction size and constant density, the elastic modulus decreases due to the weaker mechanical properties of individual carbon grains.

The stress–strain state simulations of GF samples indicated that the stress–strain states differ in both quantitative and qualitative values between isotropic and anisotropic material models. The isotropic model showed lower maximum von Mises stresses, while the anisotropic model had higher maximum shear stress tensor values. Young’s moduli determined under normal load suggested that the material behavior under strain can be described by a hyper-elastic model.

Structural analysis showed that plastic deformation under friction impacts GF from the 40–80 μm fraction more significantly, reducing the coherent-scattering region size from 28 nm to 24 nm due to higher macrostresses. For original GF samples and those that underwent static friction from the 160–200 μm and >500 μm fractions, the coherent-scattering region size increased with density from 18 nm to 22 nm, influenced by particle repositioning during rolling. The misorientation angles for GF from the 160–200 μm and >500 μm fractions decreased from 30° to 27° along RD. During friction testing along RD at 0.1 Hz and 1 Hz, no significant changes in the coherent-scattering region were observed due to the combined effects of abrasion and particle repositioning.

Microstrains and macrostresses along RD showed an overall increasing trend with density for all fractions, with values rising under higher loads due to friction. Increased values of both stress types suggest a higher degree of accumulated deformation in the samples, which likely contributes to their behavior during friction testing. This phenomenon is evident from the correlation between the observed stress levels and the roughness and friction coefficient data of the tested samples. Specifically, higher stresses are associated with greater surface roughness and varying friction coefficients, indicating that accumulated deformation influences these tribological properties. Consequently, the relationship between stress, surface roughness, and friction coefficient underscores the impact of residual deformation on the performance of the samples under frictional conditions. This aspect is subject to further and more in-depth discussion in the subsequent research papers.

Also, these findings may influence the strength characteristics of GF and will be explored further in future studies, specifically focusing on the properties of GF-based products like seals and flange gaskets.

## Figures and Tables

**Figure 1 nanomaterials-14-01499-f001:**
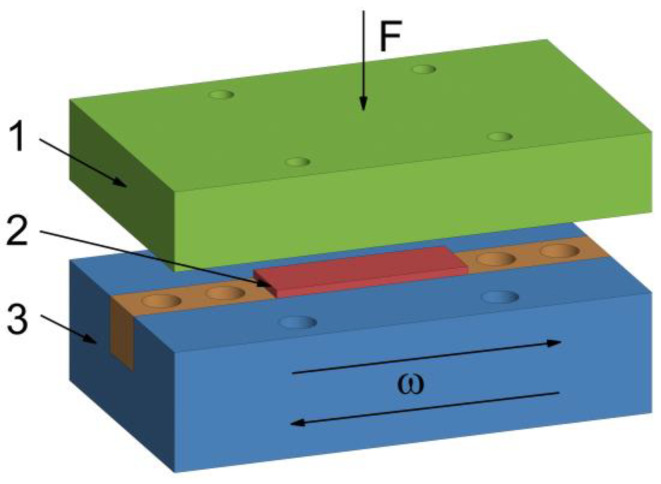
Schematic diagram of the experiment: 1—steel counterbody (also green), 2—GF sample (also red), 3—a holder (also blue), orange—fasteners, *F*—normal load, *ω*—the frequency ofreciprocating motion.

**Figure 2 nanomaterials-14-01499-f002:**
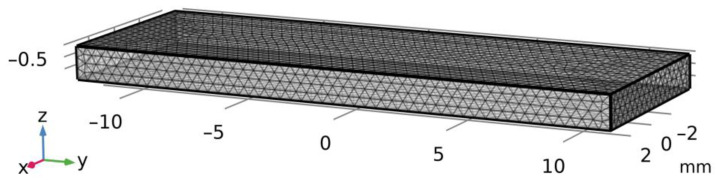
Geometry and computational grid for normal load.

**Figure 3 nanomaterials-14-01499-f003:**
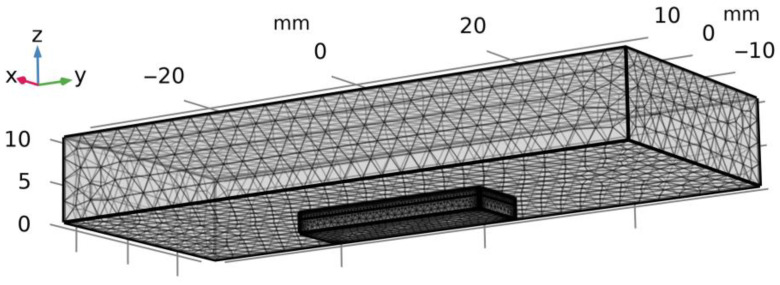
Geometry and computational grid for the tangential loading.

**Figure 4 nanomaterials-14-01499-f004:**
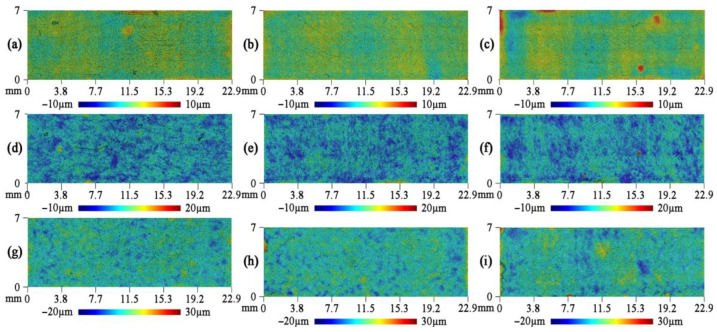
Topography of the original GF surfaces for the density of 1.0 g/cm^3^ (**a**,**d**,**g**), 1.3 g/cm^3^ (**b**,**e**,**h**), 1.6 g/cm^3^ (**c**,**f**,**i**) by the fractions of 40–80 μm (**a**–**c**), 160–200 μsam (**d**–**f**), and >500 μm (**g**–**i**).

**Figure 5 nanomaterials-14-01499-f005:**
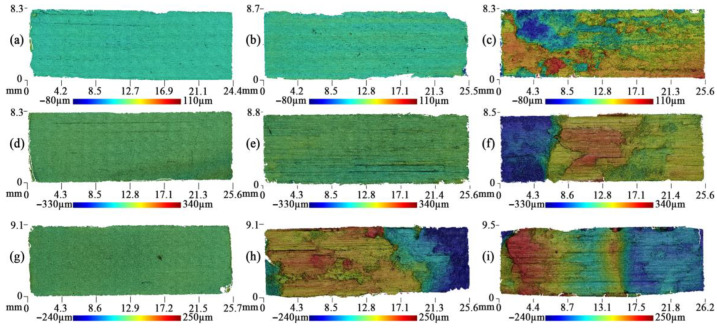
Topography of the GF surfaces for the density of 1.0 g/cm^3^ (**a**–**c**), 1.3 g/cm^3^ (**d**–**f**), and 1.6 g/cm^3^ (**g**–**i**) obtained from the fractions of 40–80 μm following the experimental studies at the sliding velocity of 1 μm/s (**a**,**d**,**g**), frequency 0.1 Hz (**b**,**d**,**h**), and 1 Hz (**c**,**f**,**i**).

**Figure 6 nanomaterials-14-01499-f006:**
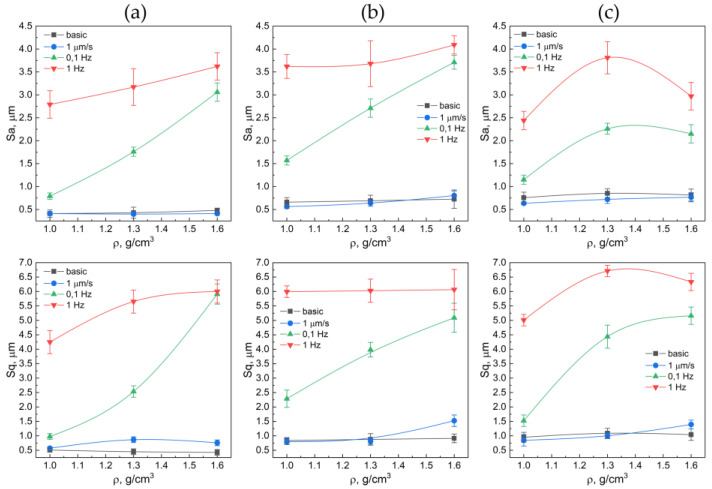
Relationship between the value of average roughness and the density in the graphite foils by the fractions of 40–80 μm (**a**), 160–200 μm (**b**), >500 μm (**c**) before (the black lines) and after (the colored lines) the frictions testings.

**Figure 7 nanomaterials-14-01499-f007:**
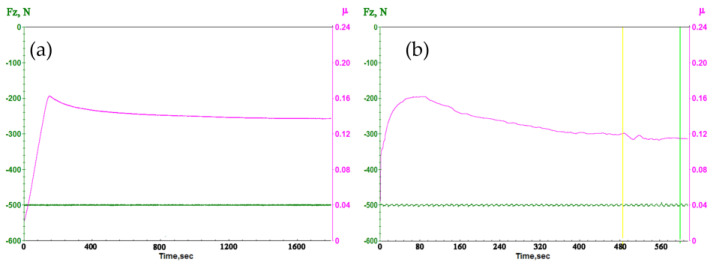
A typical view of recording static (**a**) and dynamic (**b**) friction coefficients on a UMT-3MT laboratory tribometer.

**Figure 8 nanomaterials-14-01499-f008:**
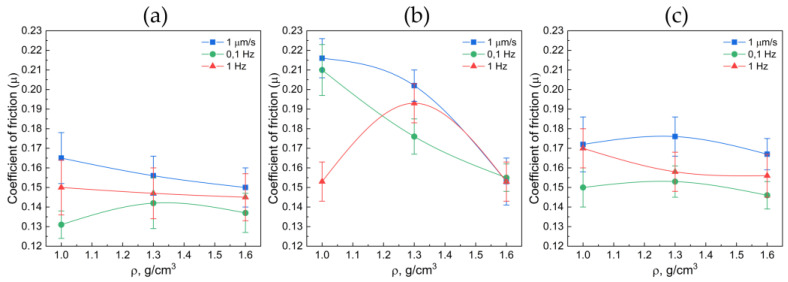
Relationship between the static (the blue curves) and dynamic (the green and red curves) friction coefficient and the density in the graphite foils by the fractions of 40–80 μm (**a**), 160–200 μm (**b**), >500 μm (**c**).

**Figure 9 nanomaterials-14-01499-f009:**
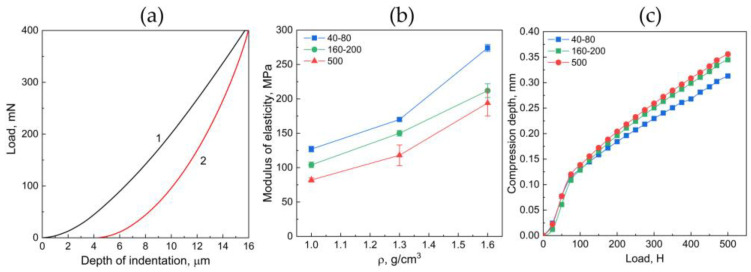
(**a**) Typical case of the relationship between load (1) and unload (2) and the penetration depth for GF material; (**b**) the elastic modulus of GF in relation to their density and fractional composition; (**c**) relationship between the compression depth and the applied load for GF, with the density being 1.0 g/cm^3^ by the fraction 40–80 μm (the black curve), 160–200 μm (the red curve), >500 μm (the green curve).

**Figure 10 nanomaterials-14-01499-f010:**
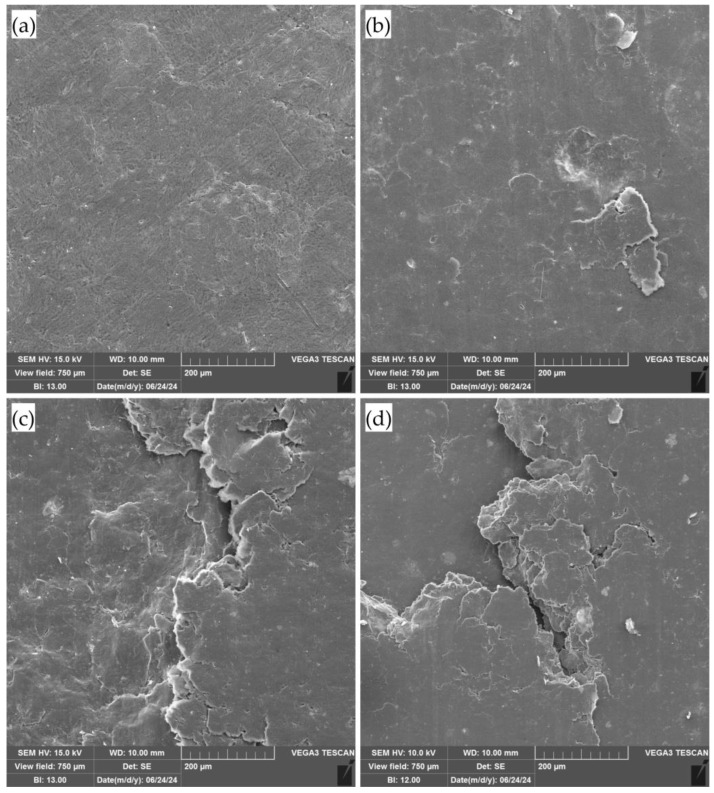
Images of the graphite foil from the 40–80 μm fraction in the original state (**a**), after the testings of static (**b**) and dynamic frictions at 0.1 Hz (**c**) and 1.0 Hz (**d**).

**Figure 11 nanomaterials-14-01499-f011:**
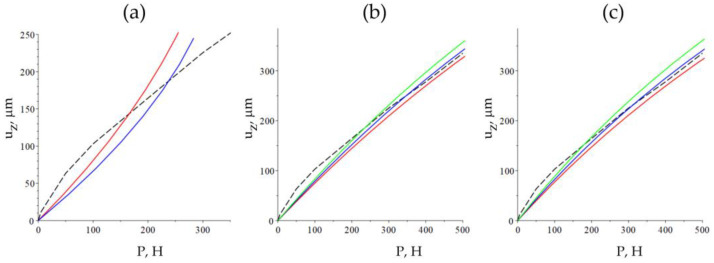
Dependences of the vertical displacement of the sample (μm) on the load *P* (N), obtained numerically (the continuous lines) and experimentally (the dashed lines); (**a**): linear elasticity, *E* = 10 MPa (the red line—ν=0.1, and the blue line—ν=0.3), (**b**): hyper-elasticity, ν=0.1 (the red line—*E* = 9.5 MPa, and the blue line—*E* = 9.0 MPa, the green line—*E* = 8.5 MPa), (**c**): hyper-elasticity, ν=0.3 (the red line—*E* = 7.5 MPa, the blue line—*E* = 7 MPa, the green line—*E* = 6.5 MPa).

**Figure 12 nanomaterials-14-01499-f012:**
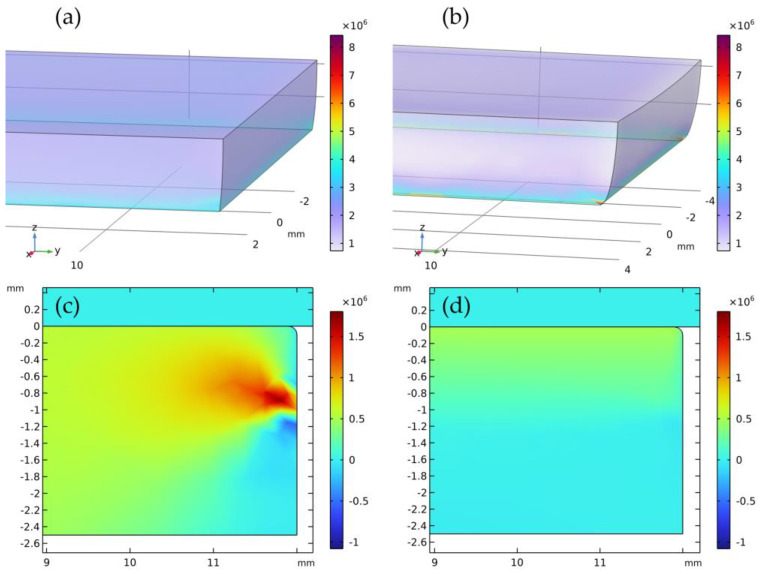
The outcome of the numerical modeling for vertical displacement at 350 μm (von Mises stress distribution over the sample volume, Pa); (**a**) ν=0.1, (**b**) ν=0.3, and distribution of the stress tensor component $\tau_{yz}$, Pa, in the bar following application of the tangential forces in the middle plane *Oyz*; (**c**) isotropic material, (**d**) anisotropic material.

**Figure 13 nanomaterials-14-01499-f013:**
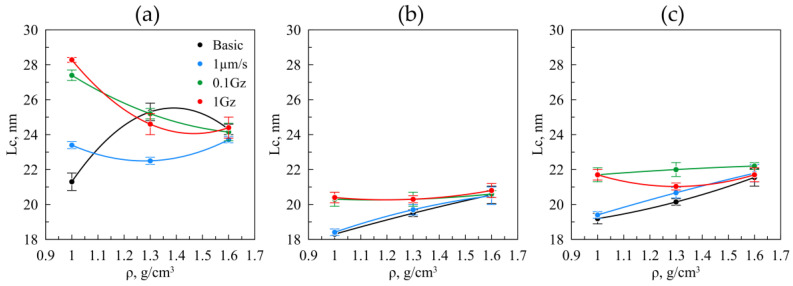
The size of the coherent–scattering region in relation to density in the graphite foils, by the fractions of 40–80 μm (**a**), 160–200 μm (**b**), >500 μm (**c**), before (the black lines) and after (the colored lines) the friction testings.

**Figure 14 nanomaterials-14-01499-f014:**
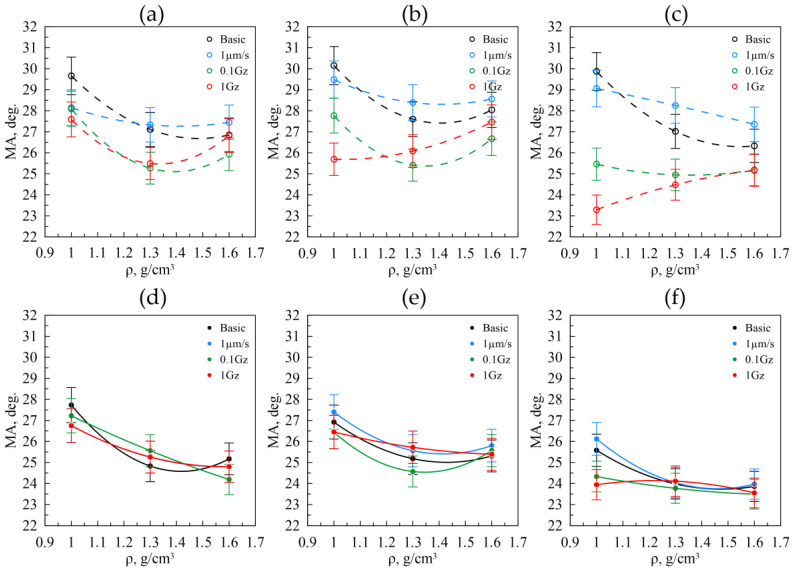
The misorientation angle size for the nanocrystallites in relation to the density in the graphite foils by the fractions of 40–80 μm (**a**), 160–200 μm (**b**), >500 μm (**c**) in RD, and from similar fractions in TD (**d**–**f**) before (the black lines) and after (the colored lines) the friction testings.

**Figure 15 nanomaterials-14-01499-f015:**
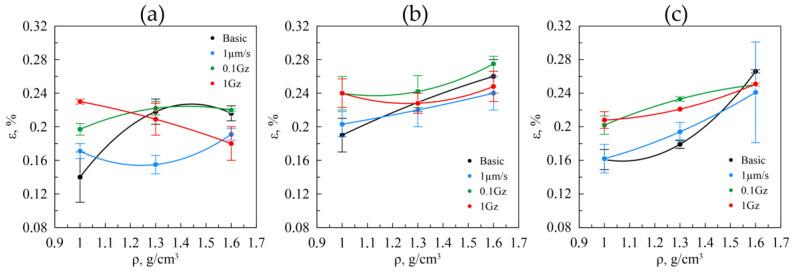
The microstrain values in the nanocrystallites in relation to the density in the graphite foils by the fractions of 40–80 μm (**a**), 160–200 μm (**b**), >500 μm (**c**) before (the black lines) and after (the colored lines) the friction testings.

**Figure 16 nanomaterials-14-01499-f016:**
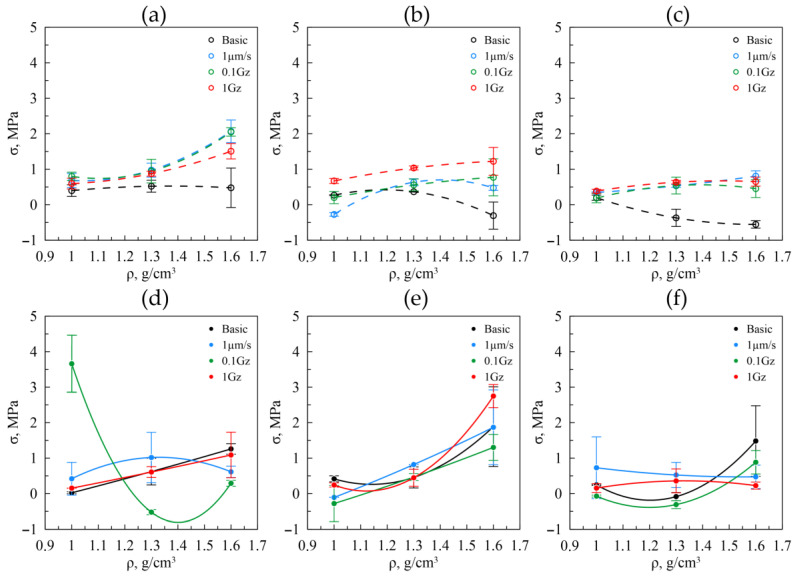
The macrostrains values in relation to the density in the graphite foils by the fractions of 40–80 μm (**a**), 160–200 μm (**b**), >500 μm (**c**) in RD, and from similar fractions in TD (**d**–**f**) before (the black lines) and after (the colored lines) the friction tests.

## Data Availability

The data presented in this study are available upon request from the corresponding author and in Appendix A.

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
