# Peer review of "Tribological Properties of Nitrate Graphite Foils"

_nanomaterials, 2024, doi:10.3390/nano14181499_

Round 1

Reviewer 1 Report

Comments and Suggestions for Authors

This paper is devoted to the examination of the coefficient of friction and surface roughness of graphite foils with different densities, obtained from the natural purified graphite. The finite element-based numerical model for the stress-strain state of graphite foils under friction for the isotropic and anisotropic scenarios was studied. The static and the dynamic friction coefficients were determined. Elastic characteristics of graphite foils were determined using the method of the instrumental indentation. Scanning electron microscopy of the samples, structural characteristics and microstrains in the nanocrystallites, misorientation angles of the nanocrystallites and residual macrostrains were also studied. I read this article with interest and the following comments come to mind:

Abstract: Numerical modeling was not mentioned.

Novelty: The literature review may not be comprehensive enough, failing to adequately showcase the background and significance of the current research. The Introduction section does not present the novelty and advantages of this work compared to other previous works. Furthermore, no review of previous studies on tribological studies of graphite foils is provided.

The details and equipment for rolling low-density GF-blanks have not been specified.

Materials and Methods: Please add the type of equipment and all devices used in the investigations by specifying the type and manufacturer.

Section 2.2: The active friction area is not given. It is impossible to estimate the nominal pressures in the contact zone.

Section 2.2: Why was a pressing force of 500 N used? Does this value correspond to real operating processes?

Section 2.2: What was the thickness of the samples?

Section 2.2: The material grade of the countersamples is not specified. The main surface roughness parameters of the countersamples should also be specified. Methods should be described with sufficient detail to allow others to replicate and build on published results.

Section 2.2: It is not stated how the friction coefficient values were determined. This is rather obvious, but formally the article should be readable for readers not familiar with the topic.

Section 2.2: It is desirable to present example curves of friction coefficient changes during the friction test for static and dynamic friction conditions.

Section 2.2: Friction tests are subject to large scatter. Therefore, they must be performed in replicates. Unless otherwise specified, the tests were performed once. Statistical analysis of the results must be performed.

I don't understand what the authors wanted to prove by conducting numerical simulations. First of all, the authors assumed that the surface roughness was not taken into consideration. Surface roughness is of fundamental importance during friction and makes the real contact area much smaller than the nominal contact area. This difference significantly affects the stress distribution. The next assumption of the friction coefficient f = 0.2 without any justification is enigmatic. Why not f = 0.8? Why not f = 0.01? As the title suggests, the article concerns Graphite Foils, and in the numerical model the sample was 2.5 mm thick. The next defect is the lack of mesh sensitivity analysis.

The graphical presentation of the sample surface topography in Figs. 4 and 5 has no practical meaning because the legends in these figures are blurred.

Analysis of the topography of samples simultaneously using the parameters Sa and Sq will not provide different qualitative results. Due to the way of determination, the parameters Sa and Sq are strongly correlated. This is commonly known. This is confirmed by similar relationships in Fig. 6a,b,c and 6d,e,f. Therefore, reliable conclusions can be obtained by analyzing only the parameter Sa or only the parameter Sq. Furthermore, Sku and Ssk are included among the parameters defining the surface topography after the friction process.

type line: ‘Fig. 8-c demonstrates the dependences of the compression depth for the GF sample on the applied load. The curves as obtained exhibit the nonlinear behaviour of this material.’ During indentation using ball indenter, the contact area (and consequently depth) changes nonlinearly with load. This applies to both elastic and elastic-plastic regimes. However, the question is: What is the physical justification that test material 'exhibit the nonlinear behaviour'? A material determination by linear stress-strain behavior will also exhibit a nonlinear relationship between load and compression depth (Fig. 8c), as explained above.

The article contains many studies that are briefly discussed in many subsections. For example, 'SEM study of the GF samples' (section 3.4) is commented on with two sentences. The mechanisms of surface wear due to friction process are not presented in detail. The same is true in other subsections. Only what was found is presented, but without a detailed discussion of why such results were obtained. In general, it would be a better article if the authors focused only on presenting the most important results discussed in detail. According to the title, the authors should focus only on Tribological Properties of Nitrate Graphite Foils.

It has not been determined how structural changes in GF affect the tribological properties of the graphite foils. The effect of microstrains and residual macrostresses in GF on the tribological properties has also not been explained.

The discussion of experimental results may not be deep enough, lacking a thorough explanation of how different parameters influence the coefficient of friction and surface roughness of samples. In order to confirm the physical relationships, the results should also be discussed with the results of other authors.

Reviewer 2 Report

Comments and Suggestions for Authors

The English is quite poor i.e. “ Obtained were the results of the surface..” this is just an example of problematic expression. Many other were found the same

The abstract as now look rather a summary and not appropriate abstract; to much info in and difficult to follow

There is reduced scientific novelty presented now in this work, as well as in the introduction.

You said “(Fig. 1-1) “ however there was presented only figure 1

In the section “2.2 Methodology for the tribological research studies “ was presented couple of parameters – do they have any physical meaning or just used because of constraint of machine ?

Figure 2 missed boundary conditions  

Figure 4 requires a proper scale bar/ the same apply for figure 5

Not clear where is the delamination in “(Fig. 5, i) delamination of GF for the densities” – I refer here that in the Figure 5 I is no sign of above claim

The conclusion are too lengthy

Comments on the Quality of English Language

The English is very difficult to understand/incomprehensible.

Reviewer 3 Report

Comments and Suggestions for Authors

The authors studied the tribological properties of GF using tribometer and FEM.  They find that friction occurring during cold rolling had similar effects, while weak loads do not cause significant defects. But the dynamic friction has a profound effect.

This is a good study and can be accepted after revision.

How many times the values are averaged to get the standard error?

Fig 4 and 5 are good. However, it is impossible to see the details of the figures. I can only see roughly some colors and roughness. These two figures have to be modified largely.

In Fig 8c why the curves turn their shape simultaneously at around 100 H?

Fig 7b, why the red curve show a very different shape than the others?

Similar questions to also Figure 15d and etc. I believe, although the data is abundant, the authors should provide further discussion for it (why the data look like this and what is the possible mechanism). This is important because reporting only the data is not sufficient to be a science work.

Comments on the Quality of English Language

Minor editing of English language required.

Round 2

Reviewer 1 Report

Comments and Suggestions for Authors

The authors answered all my comments and they modified the manuscript accordingly. I am satisfied with the answers. Consequently, at this moment I consider that the paper can be published in the Nanomaterials journal.

Reviewer 2 Report

Comments and Suggestions for Authors

.

Comments on the Quality of English Language

 Extensive editing of English language required.